# Sickness Presenteeism among Employees Having Workplace Conflicts—Results from Pooled Analyses in Latvia

**DOI:** 10.3390/ijerph191710525

**Published:** 2022-08-24

**Authors:** Svetlana Lakiša, Linda Matisāne, Inese Gobiņa, Hans Orru, Ivars Vanadziņš

**Affiliations:** 1Institute for Occupational Safety and Environmental Health, Rīga Stradiņš University, Dzirciema 16, LV-1007 Rīga, Latvia; 2Department of Public Health and Epidemiology, Rīga Stradiņš University, Kronvalda Boulevard 9, LV-1010 Rīga, Latvia; 3Institute of Family Medicine and Public Health, University of Tartu, Ravila 19, 50411 Tartu, Estonia; 4Department of Occupational and Environmental Medicine, Rīga Stradiņš University, Dzirciema 16, LV-1007 Rīga, Latvia

**Keywords:** conflicts at work, psychosocial risk factors, sickness presenteeism

## Abstract

The study’s objective was to investigate the associations between workplace conflicts and self-reported sickness presenteeism defined as going to work while being ill. Cross-sectional survey data pooled from four national surveys in years 2006, 2010, 2013 and 2018 with a study sample of 6368 employees (mean age 42.9 years and 52.9% females) were used. Respondents were randomly drawn from different regions and industries; therefore, the sample is representative of the working population of Latvia. The computer-assisted personal interviewing (CAPI) method was used to collect data at respondents’ places of residence. The associations between conflicts in the workplace and presenteeism were analyzed by using binomial logistic regression and calculated as odds ratios (ORs) with 95% confidence intervals (CIs) adjusted (aOR) for gender, age, education, and survey year. On average, 11% of respondents reported sickness presenteeism during the last year. The odds of presenteeism significantly increased for all types of workplace conflicts, but most for conflicts with managers (OR = 2.84). The odds of presenteeism doubled for those reporting conflicts with other employees (OR = 2.19) and conflicts with customers (OR = 1.85). The odds of sickness presenteeism were significantly higher if the workplace conflicts occurred often (seven times for conflicts between managers and employees, and four times for conflicts with customers) and with other employees. Presenteeism frequency increased more than three times if respondents had more than two types of conflict at work. The results of this study show that having any type of conflict in the workplace significantly increases the frequency of sickness presenteeism, especially when conflicts are frequent or an employee has more types of conflicts in the workplace. The study results justify the need to implement targeted and effective workplace conflict management measures at the organizational level to decrease sickness presenteeism.

## 1. Introduction

If sickness absenteeism means absence from work due to illness, sickness presenteeism represents a phenomenon when employees attend work even when being sick [1]. This has emerged as an important organizational issue over the last decades, and there is a substantial amount of research focusing on organizational behavior, human resources, and occupational health aspects of sickness presenteeism. When an employee suffers from any type of illness, he/she has to decide on his/her behavior—to continue working despite being ill or to take a sick leave; therefore, sickness presenteeism and absenteeism are interrelated [2]. This decision is influenced by health-related and health-non-related factors that might be individual as well as organizational [3].

Poor health of the employee is a strong determinant of sickness presenteeism [4]; however, the presenteeism differs in the case of certain health conditions (allergies, gastritis, insomnia or asthma, blood pressure problems, and thyroid trouble) [3,5] and health-related causes, which are categorized as acute illnesses, recurring complaints, chronic conditions or lifestyle factors [6,7].

Health-related factors significantly explain sickness presenteeism; however, the importance of other factors should not be diminished. The decision whether to continue work while being ill or not is also affected by the degree of incapacitation, the extent to which the employee feels able to manage their work duties, and the type of performed work [7,8]. Overall, work-related factors seem to be more important than personal circumstances in determining a person’s decision to go ill to work [9,10].

Although several organizational and occupational factors have already been linked with increased sickness presenteeism, the knowledge of the effects of psychosocial climate in the workplace is insufficient [2,11]. The following work-related psychosocial features have been associated with higher sickness presenteeism: high demands, heavy workloads, time pressure, shift work, long working hours, robust attendance management policies [6], and burnout [12], as well as lack of support in a group level and negative relationships with colleagues [10]. Robust attendance management policies, where employees fear poor evaluation or disciplinary action from colleagues or management, or where employees may lose wages or even be dismissed after a threshold level of absence is reached, also lead to increased sickness presenteeism [6,13]. According to Webster et al. [13], employees might not want to put an extra burden on colleagues as there might not be a replacement available, especially in the context of conflict presence. It has been reported that the more irreplaceable an individual is at work, the higher the correlation with sickness presenteeism [14]. In addition, the lack of replacement policy results in higher job demands and higher workloads, leading to work-related stress and emotional exhaustion, which have also been positively associated with presenteeism [10,13].

Although sickness presenteeism can be fostered by the organization’s sickness attendance culture as “presentistic”, it can be forced by the management as involuntary, or by voluntary attendance [15]. Unfortunately, employers are often unaware of the consequences of the lack of a proper sickness absence management policy. The scientific evidence suggests that presenteeism cause significantly higher costs for society than does absenteeism, and strict policy regarding sickness absence does not pay off [16,17].

Presenteeism can have a negative impact also on the company and the employee him/herself. For example, sickness presenteeism causes costs related to reduced productivity of sick employees [2,12] and the risk of spreading infectious diseases in the workplace, resulting in a burden of illness among others in the workplace [18,19]. In addition, long-term effects of sickness presenteeism can impact the health and wellbeing of the employee and can lead to increased sickness absence in the future [20,21,22], as well as increased health-related costs [7,23]. Sickness presenteeism may initiate the beginning of a process of social decline by including job loss and reduced workability. Sickness presenteeism is a strong predictor of future poor health and can result in future sickness absence [12,24,25]. Sickness presenteeism is an ambiguous phenomenon, and it is considered undesirable, but potential positive effects of working while sick are studied as well. It incorporates variables related to the person, the work, the organization, and the environment (i.e., the societal, economic, and cultural contexts) [26].

Conflicts at work may influence presenteeism through different processes, e.g., through mental health issues such as chronic stress, depression, waning motivation, and sleep disturbances that can be triggered by different workplace conflicts and lead to presenteeism afterwards [12,27]. Conflicts at work can also cause several physical symptoms through psychosomatization, such as gastrointestinal upset, asthma, ulcers, headaches, back pain, anxiety, excessive worry, loss of concentration, irritability, hypervigilance, fatigue and sleep disturbances [27,28,29]. Earlier research has revealed that low social support at work, which exists when there are conflicts in the workplace, is a risk factor for sickness presenteeism [30]. In addition, there is evidence that the existence of conflicts can impact employees’ opportunity to negotiate a replacement at work, in turn increasing the risk of sickness presenteeism [14].

However, a positive working environment can also promote sickness presenteeism, as good cooperation, loyalty, and mutual respect among colleagues can encourage employees to decide to continue to attend work while sick [31]. There are also workplace factors that in different circumstances are either positively or negatively associated with presenteeism. For example, work-related bullying is positively connected to the phenomenon of presenteeism, but person-related bullying is negatively associated [32].

Sickness presenteeism is a complex issue, and there is still a lack of knowledge on how presenteeism is affected by conflicts in the workplace. This study attempts to analyze, discuss, and reach conclusions on this research gap by investigating the association between different types and frequencies of workplace conflicts and sickness presenteeism. We have also tried to raise awareness about differences in the impact of various types of workplace conflicts on presenteeism. Our research is helpful to the scientific community because as it extends previous knowledge on workplace psychosocial risk factors particularly focusing on workplace conflicts.

## 2. Materials and Methods

### 2.1. Study Population and Sample

A cross-sectional study design was used to evaluate the association between conflicts in the workplace and self-reported sickness presenteeism. Data from four pooled national periodic workforce surveys Work conditions and risks in Latvia conducted in 2006 [33], 2010 [34], 2013 [35] and 2018 [36] were reused for this research. These national surveys aimed to evaluate the factors related to occupational safety and health, and our research was a secondary analysis of data gathered for another reason and not previously analyzed. This approach also allowed us to increase statistical significance for less frequent variables (in our case, self-reported sickness presenteeism).

The original population of the study Work conditions and risks in Latvia was representative of the working age population of Latvia and included respondents from different groups of the working population—employees, self-employed, employees on maternity leave, etc. To obtain a homogeneous study population, our data analysis included only employees; all of the other surveyed respondents were excluded from further analysis. The study sample is described in Table 1.

Respondents were randomly drawn from all regions and different industries. The average age of studied respondents (n = 6368) was 42.9 ± 12.6 (min 16, max 80 years), 47.1% were males, and 52.9% were females.

The computer-assisted personal interviewing (CAPI) method was used to collect data by interviewers at respondents’ places of residence. Questions and answers used in the survey were the same in all four surveys to ensure comparison through the study periods.

### 2.2. Study Variables

#### 2.2.1. Outcome Variable

The outcome variable in this study was self-reported sickness presenteeism (referred to as presenteeism hereinafter) within the previous year. Presenteeism was measured by the question “Which of the following situations regarding ill-health within the previous year apply to you personally?”. Possible answers were: “I was ill and took medically certified sickness absence”; “I was ill but did not take medically certified sickness absence”; “I was ill but I went to work (worked) while being ill”, “I was not ill within the previous year”.

Respondents who answered “I was ill but I went to work (worked) while being ill” were considered sickness presentees and were included in the analysis. Respondents who answered “I was not ill within the previous year” were included in the analysis as a reference group. All of the others (medically certified or self-certified sickness absentees) were excluded from the analysis (the size of the groups is given in Table 1).

#### 2.2.2. Independent Variables

Three different types of conflict in the workplace were analyzed in association with presenteeism. These conflicts were measured with the question: “Please specify how often the following situations occur in your workplace: … conflicts between management/supervisors and employees, conflicts with other employees, and conflicts with customers”. The respondents were able to select their answers regarding the frequency of conflicts from the following options of the Likert scale: “rather often”, “sometimes”, “rarely”, and “never”. Respondents who answered “I don’t know” were excluded from the analysis.

Data on the exposed group were analyzed both separately (for frequency of conflicts) and grouped as dichotomous variables. Those respondents who selected answers “rather often”, “sometimes” or “rarely” were grouped into the exposed group, but the respondents who answered “never” were used as the reference group. To identify the association between having more than one type of conflict in the workplace and presenteeism, an additional variable was calculated based on the number of different types of conflict reported by respondents: respondents who had one, two, or three different types of conflicts in the workplace.

#### 2.2.3. Confounding Variables

The following confounding variables were included in the regression models: gender, age, education, and year of the survey. Age was grouped as follows: 18–24, 25–34, 35–44, 45–54, 55–63 and 64–80. The education level was determined as preschool or incomplete primary, primary, secondary, vocational secondary, or higher education.

### 2.3. Statistical Analysis

Survey data from four periods: 2006, 2010, 2013 and 2018 were pooled into one dataset for analysis. Descriptive frequency analyses (percentages, distribution) were used to describe the study population. The association between conflicts in the workplace and presenteeism was analyzed by using binomial logistic regression and calculated as odds ratios (ORs) with 95% confidence intervals (CIs) adjusted (aOR) for gender, age, education, and survey year.

Interactions between conflicts in the workplace and the survey year were tested in association with presenteeism. No significant interactions were found, and the interaction term was not included in the analyzed regression models.

The Spearman correlation coefficient was calculated to check multicollinearity between gender, age, and education; however, no significant multicollinearity was found. The analysis was carried out using the IBM SPSS Statistics 27 (IBM Corporation, Armonk, New York, NY, USA) software.

## 3. Results

In total, sickness presenteeism was reported by 11.0% (n = 698) of respondents, varying between 7.6% and 13.8% over the survey years (Table 1). The most reported types of conflicts in the workplace were conflicts between managers and employees (53.5%), followed by conflicts between employees (44.2%), and with customers (42.3%). One-third of respondents characterized having conflicts as “rarely”, and around one-seventh as “sometimes”. The reported prevalence of workplace conflicts occurring often was the lowest (Appendix A).

The odds of sickness presenteeism significantly increased in association with different types of conflicts (Table 2). After adjustment for gender, age, education, and survey year, the association between any frequency (often, occasionally or rarely) of workplace conflicts and sickness presenteeism remained statistically significant. Adjusted odds of presenteeism were almost three times higher for those who had conflicts with managers (aOR = 2.72). A significant association was found between presenteeism and conflicts among employees (aOR = 2.12), as well as conflicts with customers (aOR = 1.78).

The odds significantly increased if workplace conflicts were more frequent. The risk was seven times higher in the group who often had conflicts with managers (aOR = 7.18), and four times higher for those who often had conflicts with other employees or with clients (aOR = 4.22 and aOR = 3.89 respectively). Additionally, responses of occasionally or rarely having conflicts were strongly associated with sickness presenteeism and at least doubled the risk of presenteeism compared to those who never had conflicts at work. For details, see Table 2.

Every fourth respondent (23.8%) had at least one type of studied conflict at work, but some respondents had more than one type of conflict at the same time (Table 3). The more types of conflicts respondents faced in the workplace, the higher was the risk of sickness presenteeism. Compared to those who had no conflicts in the workplace, respondents who reported having all three types of conflicts (between managers and employees, between employees, and with customers) had a four times increased risk of presenteeism (aOR = 4.09). The odds were three times higher if the respondent reported two types of conflicts (aOR = 3.27), and even the existence of any type of conflict doubled the risk of presenteeism (aOR = 2.01). For details, see Table 3.

## 4. Discussion

Our findings indicate that all of the types of studied conflicts at work significantly increase the frequency of sickness presenteeism. An association with presenteeism depending on frequency and number of conflicts at work was found. The odds of presenteeism increase significantly if conflicts are more frequent or if there are multiple types of conflicts at the same time.

In this study, sickness presenteeism was reported by 11.0% of all respondents included in the analyses. The report of the sixth European Working Conditions Survey (EWCS) [37] published by the Eurofound revealed significant differences in reported sickness presenteeism between the countries (from 20% to 70%). The highest levels of sickness presenteeism (>60%) were reported in Malta, France, Denmark, and Luxembourg, but the lowest (~20%) levels were reported in Portugal, Lithuania, Poland, and Bulgaria. According to the report, the prevalence of sickness presenteeism in Latvia was approximately 33%, around three times higher than the level in our study results. The discrepancy between the results of both studies can be explained by the differences in the study sample. The study population in our research included only hired employees (employees employed by organizations, e.g., companies or state institutions), whereas the population of the Eurofound study also included self-employed persons. It has already been previously reported that the level of sickness presenteeism among the self-employed is higher than among persons who are employed by organizations [9,38]. Self-employed have often higher time demands: more working hours, work in the evenings, and in their free time, indicating that it can be difficult to stay at home because of sickness [39]. In addition, the measurement of sickness presenteeism as an outcome differed between studies. Although both studies asked about going to work while being ill during the previous 12 months, the Eurofound study had a separate question about sickness presenteeism, whereas the national surveys Work conditions and risks in Latvia had a common question regarding ill-health during the previous year with multiple response possibilities (including sickness presenteeism as one of the options). It has already been previously recognized that the definition and the measurement significantly affect the reporting of sickness presenteeism, and this is one of the major sources of bias in sickness presenteeism research [8,40].

Our study provides evidence that all types of analyzed conflicts significantly increased the frequency of sickness presenteeism. From the analyzed types of conflicts, the ones between managers and employees showed almost three times higher odds of sickness presenteeism. This is in line with the findings by Kim et al. [41] that high interpersonal conflicts at work double presenteeism risk even if adjusted for general characteristics (age, gender, marriage status, smoking and drinking habits) and work-related characteristics (permitted sick leave, tenure, position, occupation, shift-work, employment type, and weekly working hours). Furthermore, our previous study about the impact of workplace conflicts on sickness absence pointed to a strong association between these particular types of conflicts and sickness absence [42]. If the impact of those conflicts on both health behavior outcomes is compared, the odds of presenteeism are significantly higher than those of absenteeism.

The impact of workplace conflicts on sickness presenteeism is not widely studied, but we can suggest several possible mechanisms of influence of workplace conflicts on sickness presenteeism. Previous research has shown the association between psychosocial stress and increased presenteeism [10,12,30,43]; therefore, all mechanisms of conflict that increase stress can also increase sickness presenteeism. For example, in situations when employees feel support from their manager, trust, and good communication, psychological stress and presenteeism is reduced [44]. In the case of the existence of conflicts, a reverse effect can be observed—presenteeism increases. The behavior of a direct supervisor can increase the occurrence of presenteeism if the supervisor is not respectful, does not cooperate, encourage, provide feedback, or help with work [45]. In addition, psychological distress can increase because of unfair treatment by the supervisor [46], which is possible in cases of conflict. Conflicts with management can also cause fear of job loss, disproportionate penalties, and a lack of equal career opportunities. This, in turn, provokes behavior leading to presenteeism, especially, in cases of staff reductions (downsizing) or related changes or decreases in staffing numbers [47]. A high level of work support can effectively prevent the negative effects of work stress and its impact on presenteeism. Increased respect and concern for employee work stress, support from colleagues and employers, and the existence of comfortable interpersonal relationships in the workplace [48], as well as the ability of managers to resolve conflicts through discussions (without exercising only their authority or failing to put any effort into resolving conflicts) can reduce presenteeism significantly [49].

The results of this research show that conflicts with colleagues double the risk of presenteeism, and this is in addition to our previous study showing that they also increase the risk of sickness absence [42]. Similar findings were revealed in another study regarding intragroup conflicts—a positive correlation was observed both for sickness presenteeism and absenteeism [3]. Increased sickness presenteeism in those who have conflicts with colleagues can be explained by the lack of possibility of replacement [1,4,13,50], lack of support from colleagues [30], fear of negative consequences [51], and stress resulting from these conflicts [3,43,48]. Conflicts with managers or colleagues can be associated with workplace bullying, discrimination, harassment, mobbing, and bossing, as these are closely related to conflict escalation of either personal or work-related relations [30,52,53].

Our results show that any type of the studied frequent conflicts significantly increases sickness presenteeism. High frequency of conflicts could also be related to workplace bullying if we define it as situations when persons are exposed to unpleasant and negative acts or behavior at work, which repeat over time. A significant relationship between exposure to workplace bullying and sickness presenteeism has already been reported earlier [54]. The existence of frequent workplace conflicts determines the psychosocial work environment, potentially affecting employees’ concentration and causing them to expend cognitive, emotional, and behavioral resources to draw up mechanisms or worry excessively about how to reduce potential risks or protect themselves [55]. Some already-studied workplace factors that increase presenteeism can be more pronounced due to the existence of conflicts: punitive sickness monitoring and pressure from management, concerns about being dismissed or being targeted for compulsory redundancy, fear of disbelief and shaming from colleagues or management [31]. The findings by Gosselin et al. [3] revealed that employees suffering from high levels of psychological stress are among those who work despite their illness.

In the current analysis, conflicts with customers and conflicts between workers also increased the odds of presenteeism. However, unlike conflicts between employees (or with managers), conflicts with customers can be not so personal and permanent, as customers change. Associations with presenteeism can be explained with mechanisms similar to those for other types of conflicts—high levels of stress can be caused by conflicts with customers and thus increase presenteeism [3,10,13,30]. Working with customers can be emotionally exhausting and lead to burnout, which in turn promotes presenteeism [12]. It should be considered that presenteeism is higher in jobs where attendance has a great influence on other people’s (customers’) health and well-being, such as in the education sectors and care and welfare. The customers in these helping professions are school children (and their parents), patients or social care persons (and their relatives) [1], and presenteeism in such cases also can be promoted by feelings of guilt [51].

The current analysis indicated that the more types of conflicts in the workplace that employees have at the same time, the considerably higher the risk of sickness presenteeism. This can be explained by the fact that more conflicts cause higher levels of emotional stress [10,12,27], an unfavorable psychosocial environment [10], and lack of support [30,48]. The risk of presenteeism is lower in a supportive workplace—if an employee can freely communicate with superiors or colleagues, rely on colleagues to receive support in case of problems, or ask for advice on personal matters [56]. The existence of conflicts, especially of more than one type or of frequent conflicts, has counter effects. Likewise, the existence of one type of workplace conflict can provoke other types of conflicts if employees’ concentration is affected by a poor social environment (existence of conflicts with supervisors or colleagues), and if employees expend cognitive and emotional resources to avoid potential risks [55], they can make more mistakes in customer service and cause conflicts with customers.

The perception of being treated unfairly by supervisors, coworkers, or customers predicts lower job satisfaction and increased psychological distress [46]. It can also lead to interpersonal conflicts, resulting in higher negative effects such as feeling upset, nervous, scared and distressed [57], and such situations can lead to increased sickness presenteeism. However, some studies show that social support can buffer the negative effects of interpersonal conflict [57] and significantly moderate the negative relationships between unfair treatment and distress (stress-buffering) [46].

There are certain limitations of this study. First, it is a cross-sectional study based on self-reported data about sickness presenteeism during the last year and the existence of different types of conflict in the workplace. Second, this research covers only the working population of Latvia, and it is not possible to compare the results or associations with other countries. The third limitation is the lack of information regarding respondents’ general health or chronic illness conditions (to be adjusted for it). Health condition is a significant predictor of sickness presenteeism, and the literature on presenteeism has investigated its links with a large number of health risks and different health conditions [4,5,58]. The study was constructed on already existing data from four pooled national periodic workforce surveys Work conditions and risks in Latvia aimed at covering general workforce information, workplace risk factors, employment issues, and legislation concepts and did not focus on employees’ health or sickness presenteeism specifically.

The fourth limitation is the risk of recall bias in a 12-month recall period. The criteria used to define an employee as a presentee were at least one episode during the last year, and that could introduce uncertainty in presenteeism group homogeneity, which is the fifth limitation of this research. Some studies use the cut-off point of two or more presenteeism episodes; however, many studies have used at least one presenteeism episode during the last year to define an employee as a presentee [9,37,59]. Data about presenteeism episodes or duration during the previous year period are missing, which limits us to a “dose-response” analysis and in terms of making clear cross-national comparisons. The measurement of presenteeism is complex and there are many differences due to measurement and definitional issues [8]. Our results cover the pre-pandemic period, and the strict stay-at-home rules implemented by different governments to mitigate the spread of the virus also might have altered the sickness behavior of employees in cases of mild symptoms of having a cold. This is the sixth limitation of our research and points to the need for extending the analyses to also cover data gathered after 2020.

Comparison of the results of this study related to the impact of conflicts on sickness presenteeism to other research is challenging because there are a limited number of similar studies. In most cases, the impact of the workplace psychosocial environment on sickness presenteeism is studied with respect to supervisor or coworker support, the existence of bullying, mobbing or harassment in the workplace, role conflicts, or work–family conflict [30,45,54,56,60]. Therefore, the theoretical implications for the framework are the investigation of the impact of particular types of conflict, frequency, or number of conflicts in the workplace on sickness presenteeism. Our approach is innovative, as no similar studies were identified. Practical applications of this study are relevant for employers, occupational health and safety experts, human resource managers and occupational health promoters, as we found a significant impact of different workplace conflicts on sickness presenteeism. Presenteeism is an undesirable phenomenon in the workplace, our findings show the differences in odds of presenteeism depending on conflict type, frequency, and the number of conflicts, and these results can help to prioritize decisions on conflict management, which in addition to improving the working environment, can also decrease costs related to sickness presenteeism.

## 5. Conclusions

The results of this study show that having conflicts in the workplace significantly increases the frequency of sickness presenteeism. Conflicts between managers and employees are associated with the highest odds of presenteeism, followed by conflicts with employees and customers. In addition, the more types of conflicts employees face in the workplace, and the higher the frequency of these conflicts, the greater the risk of sickness presenteeism. Therefore, it is essential to design targeted conflict management programs at workplaces that improve the psychosocial working environment to reduce sickness presenteeism. Our study has an innovative research topic, as we were not able to find any other similar research performed on the association between workplace conflicts and sickness presenteeism. We believe that there is a need for extending such research to other working populations, e.g., in other countries/regions and in different types of employment. Future research is also needed to obtain a deeper understanding of associations between workplace conflicts and sickness presenteeism, promote conflict management at work, reduce sickness presenteeism, and thus also improve the business performance of companies. Research also should be conducted to assess the effects of the COVID-19 pandemic on sickness presenteeism, different workplace conflicts, and their associations.

## Figures and Tables

**Table 1 ijerph-19-10525-t001:** Descriptive data for the study population.

Survey	The Total Number of the Original Study Population, n	Number of Respondents Included in Analyses, n (%)	Number of Respondents Reporting Sickness Presenteeism, n (%)	Number of Respondents in the Reference Group, n (%)
2006	2520	1601 (63.5)	166 (10.4)	1435 (89.6)
2010	2505	1578 (63.0)	218 (13.8)	1360 (86.2)
2013	2558	1579 (61.7)	192 (12.2)	1387 (87.8)
2018	2501	1610 (64.4)	122 (7.6)	1488 (92.4)
Total	10,084	6368 (63.1)	698 (11.0)	5670 (89.0)

**Table 2 ijerph-19-10525-t002:** The odds of sickness presenteeism within the previous year in association with conflicts at work.

	Sickness Presenteeism, OR (CI 95%) ^b^, Unadjusted	Sickness Presenteeism, aOR (CI 95%) ^b^, Adjusted ^c^
Conflicts between managers and employees ^a^		
Had any frequency of conflicts between managers and employees	2.83 * (2.36–3.39)	2.72 * (2.26–3.28)
Often	6.95 * (5.00–9.67)	7.18 * (5.11–10.09)
Occasionally	3.45 * (2.75–4.33)	3.35 * (2.65–4.23)
Rarely	2.25 * (1.84–2.75)	2.18 * (1.78–2.67)
Conflicts between employees ^a^		
Had any frequency of conflicts between employees	2.24 * (1.91–2.64)	2.12 * (1.79–2.50)
Often	4.79 * (2.85–8.06)	4.22 * (2.49–7.16)
Occasionally	2.99 * (2.37–3.77)	2.77 * (2.19–3.51)
Rarely	1.95 * (1.63–2.33)	1.86 * (1.55–2.23)
Conflicts with customers ^a^		
Had any frequency of conflicts with customers	1.94 * (1.65–2.29)	1.78 * (1.50–2.10)
Often	4.30 * (3.05–6.05)	3.89 * (2.73–5.52)
Occasionally	2.20 * (1.77–2.73)	2.00 * (1.60–2.50)
Rarely	1.55 * (1.27–1.89)	1.43 * (1.16–1.75)

^a^. The reference category for conflict group is the group of respondents who did not have this particular type of conflict at work (answered “never”). ^b^. The reference category for sickness presenteeism group is a group of respondents who did not get sick in the previous year. ^c^. Adjusted for gender, age, education, and survey year. * *p* < 0.001.

**Table 3 ijerph-19-10525-t003:** The odds of sickness presenteeism within the previous year in association with more than one type of conflict at work.

	Distribution of the Number of Conflicts, % (n)	Sickness Presenteeism, OR (CI 95%) ^b^, Unadjusted	Sickness Presenteeism, aOR (CI 95%) ^b^, Adjusted ^c^
Had one type of conflict at work ^a^	23.8 (1473)	2.06 * (1.56–2.72)	2.01 * (1.52–2.67)
Had two types of conflicts at work ^a^	25.4 (1569)	3.37 * (2.60–4.37)	3.27 * (2.51–4.26)
Had three types of conflicts at work ^a^	21.7 (1340)	4.44 * (3.43–5.75)	4.09 * (3.13–5.35)
Had no conflicts at work	29.2 (1805)	1	1

^a^. The reference category for conflict groups is the group of respondents who did not have any type of conflicts at work. ^b^. The reference category for sickness presenteeism group is a group of respondents who did not get sick in the previous year. ^c^. Adjusted for gender, age, education, and survey year. * *p* < 0.001.

## Data Availability

The original datasets of the independent studies Work conditions and risks in Latvia are available upon an official data request to the relevant data owners: 2006—from the Ministry of Welfare (https://www.lm.gov.lv/en (accessed on 9 May 2022)), 2010 and 2013—from the Employers’ Confederation of Latvia (https://lddk.lv/en/ (accessed on 9 May 2022), 2020—from the State Labour Inspectorate (https://www.vdi.gov.lv/en (accessed on 9 May 2022).

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
