# Peer review of "Sickness Presenteeism among Employees Having Workplace Conflicts—Results from Pooled Analyses in Latvia"

_ijerph, 2022, doi:10.3390/ijerph191710525_

Round 1

Reviewer 1 Report

Dear authors,

I congratulate you on a well-conducted research and a well-written manuscript. The only suggestions I have are: expand Conclusion section.

  The manuscript could be "improve" in terms of English language, as well.  This mostly refers to the "Discussion" section.

Author Response

Thank you very much for reviewing our manuscript titled “Sickness presenteeism among employees having workplace conflicts – results from pooled analyses in Latvia". We appreciate your suggestions, comments, and the time you dedicated to providing feedback on our manuscript. We have revised the manuscript and highlighted the changes within the manuscript.

Sincerely, Svetlana Lakiša.

Reviewer 2 Report

The theme of this paper is very interesting and innovative, and certainly useful for society.

This paper could be interesting for researchers looking for this Journal.

However, this paper has some weaknesses.

ABSTRACT

The abstract is what attracts (or does not) the attention and interest to the article. So, it should be carefully written.

It is suggested that the authors refer to the methodology used.

It is suggested that the authors simplify the main conclusions obtained e retired all statistical data (e.g. OR=2.84, 95% CI 2.36- 193.41).

1.INTRODUCTION

An introduction should be informative and well-worded. Therefore, it is missing:

1-  present adequately the theoretical problem/formulation and the objective of the study.

2-  give clues to the discussion of the results (methodology used).

3-  present the structure of the article.

The paper mentions previous studies from 2006 [33], 2010 [34], 2013 [35] and 2018 [36] (line 113 and 114), but are not presented in the text. It is suggested that the authors include a summary of the main conclusions of each study, so that it is not necessary to read these studies to understand the article.

2. MATERIAL AND METHODS

It shall provide the necessary and sufficient information to assess how the study was conducted in order to allow its reproduction by other.

All this relevant information is missing:

1.   There is no presentation of the questionnaire, and it is not known if it was answered online or in person?

2.   It would be interesting to put the questionnaire in Appendix, so that other researchers replicate the study

3.   Nor is the information about the N and the % of each group were considered in 2.2.1. Study population and sample

4.   It is not understood which methods are used to obtain the results, i.e., explain the methodology used

3. RESULTS

This point should be articulated with the previous one, as the results presented should be supported by the descriptive methods.

The results should show the evidence of the study and should be presented according to a logical and informative and perceptible sequence.

This point should be improved.

It is suggested that it joins as Tables 2, 3 and 4 in a single.

4. DISCUSSION

The theoretical implications and possible practical applications should be discussed.

When the authors refer to "our study" it should point out that this is a study based on the reality of Latvia

Limitations - I don't understand because this point is not in the conclusions!

5. CONCLUSION

The results should be presented in a comprehensive manner, highlighting the most relevant ones and provide a summary of the text.

The originality and relevance of the results presented should be strengthened.

The limitations of the study itself should be presented and discussed here and not at a previous point.

Another limitation of this research is that the study focuses only on Latvian society.

The future investigation need referred here.

Suggestions for future research:

1. Conduct the same study in other countries/regions of the world

2. The data this study are from 2018 (before the pandemic), so it would be interesting to conduct the same study regarding what happened in the pandemic or even in the post-pandemic situation

3. To analyze the impact of this reality on the different conflicts identified in article

Author Response

(The authors gave the same response as above.)

Round 2

Reviewer 2 Report

No comments. Congratulations!